# Changes in COVID-19 Vaccine Acceptability among Parents with Children Aged 6–35 Months in China—Repeated Cross-Sectional Surveys in 2020 and 2021

**DOI:** 10.3390/vaccines11010170

**Published:** 2023-01-12

**Authors:** Kechun Zhang, Xue Liang, Karen Lau Wa Tam, Joseph Kawuki, Paul Shing-fong Chan, Siyu Chen, Yuan Fang, He Cao, Xiaofeng Zhou, Yaqi Chen, Tian Hu, Hongbiao Chen, Zixin Wang

**Affiliations:** 1Longhua District Center for Disease Control and Prevention, Shenzhen 518110, China; 2Jockey Club School of Public Health and Primary Care, Faculty of Medicine, The Chinese University of Hong Kong, Hong Kong SAR, China; 3Department of Health and Physical Education, The Education University of Hong Kong, Hong Kong SAR, China

**Keywords:** parental acceptability, COVID-19 vaccination, changes, children aged 6–35 months, repeated cross-sectional surveys, China

## Abstract

China is considering to offer COVID-19 vaccination for children aged 6–35 months. This study investigated the changes in COVID-19 vaccine acceptability and associated factors among parents with children aged 6–35 months in 2020 and 2021. Two rounds of cross-sectional online surveys were conducted among adult factory workers in Shenzhen, China. A subset of 208 (first round) and 229 (second round) parents with at least one child aged 6–35 months was included in the study. Parental acceptability of COVID-19 vaccination increased significantly from 66.8% in the first round to 79.5% in the second round (*p* = 0.01). Positive attitudes, perceived subjective norm, and perceived behavioral control were associated with higher parental acceptability in both rounds of surveys (*p* values ranged from <0.001 to 0.003). A negative association of negative attitudes with parental acceptability was observed in the second round (*p* = 0.02). No significant associations of exposure to information related to COVID-19 vaccination on social media with parental acceptability was found in either round of survey. Expanding the existing COVID-19 vaccination programs to cover children aged 6–35 months is necessary in China. Future programs should focus on modifying perceptions among parents to promote COVID-19 vaccination for children in this age group.

## 1. Introduction

Coronavirus diseases 2019 (COVID-19) is one of the major challenges in the 21st century [1]. The ongoing pandemic has resulted in global health, economic, and social crises. As of 28 November 2022, there have been 637 million confirmed cases including 6.6 million deaths worldwide [2]. In addition, some countries (e.g., China) repeatedly applied strict measures (i.e., routine COVID-19 screening, social distancing, and even lockdown of the entire city), with significant and long-lasting individual, social, and economic consequences [1]. In the early phase of the COVID-19 pandemic, most cases were middle-aged and older people [3,4], but as the pandemic progressed, an increasing trend in children with COVID-19 infection was observed [4]. Younger children are especially vulnerable to COVID-19 [5]. Children aged 6–35 months accounted for 3.6% of all COVID-19 cases in the United States [6], 3.1% in German [7], and 2.2% in the United Kingdom [8]. The proportion of children aged 6–35 months hospitalized due to the Omicron variant was about five times higher than when the Delta variant was the dominant strain [9]. Such a proportion was also higher than that of children aged 5–11 and 12–17 years during the same period [10]. In children, younger age was positively associated with COVID-19 mortality [11]. Therefore, preventive methods are urgent to protect younger children from COVID-19.

COVID-19 vaccination is highly effective in preventing COVID-19 and its equality should be guaranteed in populations and communities at risk of infection [12,13,14]. Vaccine efficacy was also evaluated in children aged between six months and five years and similar robust neutralizing antibody titers were observed [15]. Compared to young adults, the vaccine immunogenicity was non-inferior in children aged between six months and five years [16]. In this group of children, most adverse effects reported were mild to moderate and serious side effects were rare [17,18]. In June 2022, the United States amended the Emergency Use Authorization to include children aged 6–35 months for COVID-19 vaccination [19]. As of 16 November 2022, 1.7 million children aged between six months and four years have received at least one dose of COVID-19 vaccine, representing 10% of this age group in the United States [20]. Since September 2022, Australia has offered the Moderna vaccine to children aged between six months and four years [21]. COVID-19 vaccine was also approved for children aged six months to five years in Canada in July 2022 [22] and for those aged 6–35 months in Hong Kong in August 2022 [23]. China approved two inactivated COVID-19 vaccines (Sinopharm and SinoVac-CoronaVac) for children aged 3–17 years in July and October 2021 [24], and is considering to offer COVID-19 vaccination to children aged 6–35 months.

Sufficient vaccination coverage is conditioned by the people’s acceptance of these vaccines. However, providing children with COVID-19 vaccination is challenging due to low parental acceptability, especially for those of younger age [25]. Studies about parental acceptability of COVID-19 vaccination for children aged 6–35 months are still limited [26,27]. One cross-sectional study conducted in the United States showed that only 40% of parents were willing to vaccinate their children aged between six months and four years against COVID-19 [26]. A similar level of COVID-19 vaccination acceptability was found among parents with children under 18 months in the United Kingdom [27]. Routine childhood vaccination, receiving the seasonal influenza vaccine, perceived severity of COVID-19 in children, vaccine safety, and effectiveness were associated with higher vaccine acceptability [26]. Common concerns, such as vaccine safety, effectiveness, and rapid vaccine development, were negatively associated with vaccine acceptability [27]. In China, most studies investigating parental vaccine acceptability for children focused on children aged 6–18 years [28,29,30,31,32,33,34,35,36,37,38,39,40,41]. Our previous work investigated the changes in parental COVID-19 vaccine hesitancy for children aged 3–17 years [41]. Since the recommendation and policy for COVID-19 vaccination are very different for children aged 3–17 years and those aged 6–35 months, we expect the level of parental acceptability and determinants would be different between parents of older and younger children. According to the principle of social marketing, health promotion should be tailored to the needs of the target population [42]. Therefore, it is necessary to evaluate the parental acceptability of COVID-19 vaccination for children aged 6–35 months. This study could address the knowledge gap and inform the design of interventions/programs promoting COVID-19 vaccination among children aged 6–35 months.

The objectives of this study included: (1) to investigate the changes in parental acceptability of COVID-19 vaccination for children aged 6–35 months in 2020 and 2021 among factory workers in Shenzhen, China, and (2) to investigate factors associated with parental acceptability of COVID-19 vaccination at each time point. In the first round of survey, COVID-19 vaccination was not yet available in China. In the second round of survey, COVID-19 vaccination was available for children aged three years or above but was not yet available for children under three years. We hypothesized that parental vaccine acceptability would increase over time, and associated factors would differ at these two-time points.

## 2. Materials and Methods

### 2.1. Study Design

Two rounds of cross-sectional online surveys were conducted to explore COVID-19 vaccine acceptability among adult factory workers in Shenzhen Municipality, China from 1 to 7 September 2020 and 26 to 31 October 2021, respectively [41,43,44]. The study sites, sampling, and data collection of the two rounds were the same [41,43,44]. Shenzhen Municipality is a major and special economic zone in China and most factories were built in Longhua district, with 1517 factories and over one million factory workers in 2020 [45]. COVID-19 vaccination was available for children aged 12–17 in July 2021 [41] and 3–11 years in October 2021 [24].

### 2.2. Participants and Data Collection

Full-time factory workers aged 18 years or above in Shenzhen Municipality were recruited in the two rounds of surveys. We extracted data of participants having children aged 6–35 months for this study. Details of data collection have been described previously [41,43,44]. Factory workers in Shenzhen are required to receive physical examination every 12 months. Such physical examination is required to renew their working permit/contract. All organizations in Longhua District providing annual physical examination to factory workers, including three public hospitals, two private hospitals, and the CDC, were selected for recruitment of participants in both rounds. Fieldworkers approached all workers attending the sites for physical examination during the study period and obtained informed consent from potential participants. Since the interval between the first and second round was 14 months, most factory workers who participated in the first round should have completed the physical examination before the start of the second round. Therefore, our participants were likely to comprise two random samples derived from about one million factory workers. The chance of having the same participant completing both surveys was very low and not of practical concern.

A common encrypted online survey platform, Questionnaire Star, was used to collect information. Quick response (QR) codes were provided to access the online questionnaire and each mobile device was only allowed once for the questionnaire response to avoid duplication. The survey included about 20 items and took around 20 min to complete. A completeness check was performed before participants submitted their questionnaires. An e-coupon with 10 RMB (1.5 USD) was sent to participants once completion. All data was stored on the online survey platform server, and only the corresponding author had access to the database. Longhua District CDC provided the ethics approval (references 2020001 and 2021015).

### 2.3. Measures

#### 2.3.1. Design of the Questionnaire

A panel, including one member of CDC staff, two public health researchers, a health psychologist, and a factory worker, developed the questionnaire. A pilot test was conducted to assess the clarity and readability among 10 factory workers, who did not participate in the actual survey. The questionnaire was revised and finalized based on the workers’ comments.

#### 2.3.2. Background Characteristics

Demographic information, including age, sex, relationship status, education, monthly personal income, type of work, and age of the child were collected. Compliance with personal preventive behaviors was estimated using validated tools [46,47,48], including frequency of face mask wearing when having close contact with others in the workplace and other public spaces and hand sanitizing after returning from public spaces or touching public installations (every time, often, sometimes and never). Self-reported avoiding of social and meal gathering with people who did not live together and in crowded places in the last month was also measured.

#### 2.3.3. COVID-19 Vaccine Acceptability for Their Children

Vaccine acceptability in both rounds of surveys was assessed by asking, “What is your likelihood of having your child take up free COVID-19 vaccination provided by the government?” (Response categories: 1 = very unlikely, 2 = unlikely, 3 = neutral, 4 = likely and 5 = very likely). Parental acceptability was defined as “very likely” or “likely” in both rounds. The same definition of parental acceptability was used in other published studies [29,49].

#### 2.3.4. Individual Level Factors: Parental Hesitancy to Receive a COVID-19 Vaccination and Attitudes toward COVID-19 Vaccination for Their Children

In both rounds, parents who had not received COVID-19 vaccination were asked about their likelihood to take up such vaccination in future (response categories: 1 = very unlikely, 2 = unlikely, 3 = neutral, 4 = likely and 5 = very likely). Unvaccinated parents who were very unlikely/unlikely/neutral to take up COVID-19 vaccination future were defined as having parental vaccine hesitancy.

Items/scales on attitudes toward COVID-19 vaccination were constructed in our previous study, which was one of the first studies looking at parental attitudes toward COVID-19 vaccination. Since there was no validated questionnaires on this topic in early 2020, the items and scales were self-constructed based on those from previous studies on parental acceptability of human papillomavirus vaccination in China [50]. Positive and negative attitudes toward COVID-19 vaccination were measured by two scales. The Cronbach’s alpha of these two scales in the original study was 0.71 and 0.64, respectively. The internal reliability was acceptable. In the original study, we conducted exploratory factor analysis for these two scales. The scales had single factors explaining 64.0% and 56.6% of the total variance (Appendix A). Perceived subjective norm and perceived behavioral control were assessed by asking “Your family member would support you in having your child take up COVID-19 vaccination” and “Having the child receive COVID-19 vaccination is easy for you if you want them to”, respectively. Disagree, neural and agree were rated for each item.

#### 2.3.5. Interpersonal Level Factors: Influence of Social Media

In both rounds, participants were asked about the frequency of exposure to experiences related to COVID-19 vaccination on social media (such as WeChat and Weibo), shared by recipients in the last month. In the second round, four other types of information on social media were collected, including (1) the COVID-19 pandemic is not under control in some countries after scaling up COVID-19 vaccination, (2) infectiousness and harms of the variants concern of COVID-19, (3) outbreak caused by variants concern of COVID-19 in some places of China, and (4) people contract COVID-19 after receiving primary series of COVID-19 vaccines. Each item was rated from 1 = almost never, 2 = seldom, 3 = sometimes, to 4 = always.

### 2.4. Sample Size Planning

The original study had two rounds of cross-sectional online surveys looking at COVID-19 vaccination uptake and attitudes among adult factory workers in Shenzhen, China. The target sample size was 2000 for the original study in both rounds, which has been explained in previous papers [43,44]. This study was based on a sub-sample of these participants who had at least one child aged 6–35 months. Our reference population was factory workers having children aged 6–35 months in Shenzhen, China. Assuming that approximately 10% of the participants had a child aged 6–35 months (*n* = 200), we estimated about 60–70% of parents showed vaccine acceptability for their children in the first round. The smallest between-round difference in vaccine acceptability of 9.9% could be detected (power = 0.80, alpha = 0.05; PASS 11.0, NCSS, LLC, Kaysville, UT, USA).

### 2.5. Statistical Analyses

Differences in background characteristics, parental acceptability, attitudes toward COVID-19 vaccination and social media influence between participants in the first and second round of survey were compared using Chi-square tests (for categorical variables) or independent-sample t-tests (for continuous variables). Unadjusted *p* values were obtained. Using parental acceptability, attitudes toward COVID-19 vaccination, and social media influence as dependent variables, round of survey as independent variable (second versus first), and adjusting for significant background characteristics with between-group differences, adjusted *p* values were obtained by logistic regression models (for binary dependent variables) or linear regression models (for continuous dependent variables). Regarding the effect sizes of the between-round comparisons, Cohen’s d was calculated for continuous variables and Cohen’s h was calculated for proportions. For parents in the same round of the survey, Logistic regression models were used to assess the association of vaccine acceptability with background characteristics, attitudes toward COVID-19 vaccination, and influence of social media, giving odds ratio (ORs) and 95% confidence interval (CI). Statistical analyses were conducted using R software (version 4.1.2, ST Louis, MO, USA). All tests were two-sided with *p* < 0.05 as statistically significant.

## 3. Results

### 3.1. Background Characteristics of the Participants

A total of 2653 and 3060 eligible factory workers were approached in the first and second rounds of surveys, 600 and 434 refused to join the study due to a lack of time or other logistical reasons, and 2053 and 2626 completed the surveys with the response rate being 77.3% and 85.8%, respectively. Among these participants, 208 and 229 with at least one child aged 6–35 months were included in the analyses.

Table 1 showed that the mean age of parents was 30.9 and 33.1 years in the first and second round, respectively. The majority of them were female (first round: 57.2%, second round: 57.2%), married (first round: 97.6%, second round: 97.8%), with monthly personal income between 3000 and 6999 RMB (first round: 53.8%, second round: 60.7%), working as frontline workers (first round: 63.5%, second round: 69.9%), and with children aged 24–35 months (first round: 53.8%, second round: 52.0%). As compared with parents in the first round, those in the second round were older (*p* < 0.001) and had higher education levels (*p* = 0.02).

### 3.2. Changes in Parental Acceptability and Attitudes toward Children’s COVID-19 Vaccination

Parental acceptability of COVID-19 vaccination increased significantly from 66.8% in the first round to 79.5% in the second round (*p* = 0.01). Significant increases were also found in positive attitudes, negative attitudes, perceived subjective norm, and perceived behavioral control related to children’s COVID-19 vaccination (*p* values ranged from <0.001 to 0.04). More parents in the second round were exposed to testimonials given by recipients of COVID-19 vaccination on social media compared to those in the first round (*p* < 0.001) (Table 2).

### 3.3. Factors Associated with Parental Acceptability of COVID-19 Vaccination

In the first round, better compliance with physical distancing behaviors was associated with higher parental acceptability. In the second round, being female, and having better compliance in terms of face mask wearing in the workplace and hand hygiene were associated with higher parental acceptability. Higher education level was negatively associated with parental acceptability (Table 3).

After adjusting for the above significant background characteristics, positive attitudes (adjusted odds ratio (AOR): 1.86 and 1.99, *p* < 0.001), perceived subjective norm (AOR: 4.69 and 8.33, *p* < 0.001), and perceived behavioral control (AOR: 2.02 and 3.09, *p* = 0.003 and <0.001) were associated with higher parental acceptability for children in the first and second rounds, respectively. Parental hesitancy to receive COVID-19 vaccination was negatively associated with vaccine acceptability for children in the first round (AOR: 0.25, *p* < 0.001). In the second round, negative attitude was associated with lower parental acceptability (AOR: 0.80, *p* = 0.02). The association between frequency of information exposure on social media and the dependent variable was not statistically significant in either round (Table 4). Regarding the associations between individual items of the Positive/Negative Attitude Scale and the dependent variable, belief that COVID-19 vaccination is highly effective in protecting your child from COVID-19 was associated with higher parental acceptability in both rounds of surveys. Perceive adequate supply of COVID-19 vaccines was associated with higher parental acceptability in the first round of surveys, but not in the second (Appendix B).

## 4. Discussion

To our knowledge, this was the first study to monitor the changes in COVID-19 vaccine acceptability among parents with children aged 6–35 months. Factors at both individual and interpersonal levels were considered, which provided a more comprehensive understanding about the determinants of parental acceptability. The findings enriched international literature about parental acceptability of COVID-19 vaccination for children under the age of three years. The findings also had some practical implications to inform development of vaccination programs and related health promotion for this group of children. Currently, the risk of COVID-19 infection under the age of three years increased dramatically after the country changed its zero COVID policy. It was expected that the fear about COVID-19 and its consequences would be a major stressor for parents of young children [1]. Rolling out a vaccination program for young children is the first method to mitigate the impacts of pandemic on parents.

Parental acceptability of COVID-19 vaccination for children increased from 66.8% in the first round to 79.5% in the second round, possibly due to the increasing coverage of COVID-19 vaccination in adults. Most parents had already received the vaccine when the second round of survey was conducted. As of 3 December 2022, 90.7% of the entire population in China had received primary series of COVID-19 vaccine [51]. In addition, the recommendation of COVID-19 vaccine for children aged 3–17 years [52] may be another reason for the increased acceptability. The coverage of COVID-19 vaccination among children in China has been increasing rapidly without serious safety concerns [53]. The levels of parental acceptability in both rounds were higher than that reported in the United States (39.7%) [26] and the United Kingdom (48.2%) [27]. However, such levels of parental acceptability among our participants were lower when compared to Chinese parents having older children [31,33,34,37,38,40,49]. Younger age of the children and the availability of vaccines might be reasons to explain such differences. As compared to other jurisdictions (i.e., United States, Australia, Canada, and Hong Kong SAR of China) [21,22,23], mainland China was late in rolling out COVID-19 vaccination for children aged below three years. China may learn from the experience and evidence of these jurisdictions and expand the existing vaccination programs to children aged 6–35 months. This is especially important for protecting these young children after the country decided to change its zero COVID policy.

Compared to the first round, significant increases were observed in attitudes favoring COVID-19 vaccination for children, including positive attitudes, perceived subjective norm, and perceived behavioral control in the second round. The health promotion conducted by governments, availability of COVID-19 vaccine for children aged 3–17 years, and associated vaccination experiences from older children may explain the observed changes. Positive attitudes were associated with higher parental COVID-19 vaccine acceptability for children, which was consistent with previous studies [27,29,31,33,41]. When promoting COVID-19 vaccination for children aged 6–35 months, health communication messages emphasizing importance of childhood vaccination in overall pandemic control and sufficient supply of the vaccines should be disseminated to parents. In addition, studies should obtain more data about the efficacy of inactivated COVID-19 vaccines among young children, as such information is also useful in increasing parents’ positive attitudes toward the vaccines. However, negative attitudes also increased over time, with a higher proportion of parents concerned about the short duration of protection and time constraints to take their children for COVID-19 vaccination. The COVID-19 booster dose was unavailable for children in China, which may partly explain the concern of protection duration. A booster dose was shown to be effective in a long-time vaccine efficacy trial in children aged 5–15 years from the United States [54]. Kindergarten-based COVID-19 vaccine programs may partly solve the time concern among working parents. Such an approach can also increase perceived behavioral control related to children’s COVID-19 vaccination among parents, which was a significant determinant in both rounds of surveys. Moreover, perceived support regarding children’s vaccination increased over time and was associated with higher parental acceptability at both rounds. Future programs promoting COVID-19 vaccination among young children should encourage parents to communicate with other family members to obtain more support from these significant others.

Some non-significant findings also provided insights for planning future interventions. In contrast to our hypothesis, exposure to information related to COVID-19 vaccination on social media was not a significant determinant of parental acceptability in either round of survey. Therefore, future programs promoting COVID-19 vaccination among children aged 6–35 months should focus on modifying perceptions among parents.

Our study had several limitations. First, our findings are most applicable to the time when zero COVID policy was implemented in China. When China started to relieve its strict COVID-19 control measures in December 2022, the number of new cases increased dramatically. Parents may perceive a much stronger need to vaccinate their children against COVID-19. Second, participants referred to the child whose birthday was closest to the survey date when answering questions related to parental acceptability if they had more than one child under the age of 18 years. Therefore, we did not collect information about their other children’s COVID-19 vaccination status. Other children’s vaccination status may represent an important determinant of parental acceptability for their younger children. Third, the history of COVID-19 infection was not collected in our surveys, and an Italian study showed that parents with children previously experiencing COVID-19 were more likely to support the vaccination [55]. However, as the daily confirmed cases were very low during the two surveys due to the zero COVID policy in China, the COVID-19 infection history would not substantially influence the results. Fourth, we did not ask participants in the second round whether they attended the first round of survey. In Shenzhen, factory workers are required to receive physical examination every 12 months to renew their working permit/contract. The interval between the first and second round of survey was 14 months. The chance of a participant completing both rounds of surveys was very low. Fifth, the second round did not measure awareness of childhood COVID-19 vaccination. However, the awareness should be high among parents, as in the past two years, COVID-19 vaccination for children has been strongly promoted via various social media in China. Sixth, all participants were factory workers. Failure to include parents with other occupations or those without full-time work was one major limitation of this study and limited the representativeness of our sample. All participants were recruited in Shenzhen Municipality, one of the first-batch cities for COVID-19 vaccination in China which may have higher parental vaccine acceptability than less developed regions. Generalization of the results should be taken with caution when applying the findings to other cities. Moreover, selection bias existed due to non-response. Characteristics between participants and refusals could not be compared as no information was collected from refusals. Nonetheless, our response rate in both rounds was higher than in studies with similar topics [29,30,31,34,37,38,40,41,49]. Last but not least, the study was cross-sectional and so it could not establish causal relationships.

## 5. Conclusions

This study compared parental COVID-19 vaccination acceptability for children aged 6–35 months in 2020 and 2021 among factory workers in Shenzhen, which is one of four most developed cities in China. The prevalence of parental acceptability was relatively high and had been increasing over time. Expanding the existing COVID-19 vaccination programs to cover this age group of young children is necessary in China. Future programs promoting COVID-19 vaccination for children in this age group should focus on modifying perceptions among parents, such as increasing positive attitudes, perceived subjective and perceived behavioral control, and reducing negative attitudes related to children’s vaccination.

## Figures and Tables

**Table 1 vaccines-11-00170-t001:** Background characteristics of the parents.

	Round 1 (*n* = 208)	Round 2 (*n* = 229)	*p* Values
	*n* (%)	*n* (%)	
**Sociodemographic characteristics**			
Age of the parent, years			
18–30	106 (51.0)	80 (34.9)	
31–40	97 (46.6)	124 (54.1)	
>40	5 (2.4)	25 (10.9)	<0.001
Sex			
Male	89 (42.8)	98 (42.8)	
Female	119 (57.2)	131 (57.2)	1.00
Relationship status			
Married	203 (97.6)	224 (97.8)	
Single or divorced	3 (1.4)	5 (2.2)	
Having a stable partner	2 (1.0)	0 (0.0)	0.28
Education			
Senior high or below	127 (61.1)	113 (49.3)	
College and above	81 (38.9)	116 (50.7)	0.02
Monthly personal income, RMB (USD)			
<3000 (462)	49 (23.6)	35 (15.3)	
3000–6999 (462–1077)	112 (53.8)	139 (60.7)	
≥7000 (1078)	47 (22.6)	55 (24.0)	0.09
Type of work			
Frontline workers	132 (63.5)	160 (69.9)	
Management staff	76 (36.5)	69 (30.1)	0.19
Age of the child, months			
24–35	112 (53.8)	119 (52.0)	
12–23	91 (43.8)	95 (41.5)	
6–11	5 (2.4)	15 (6.6)	0.12
**Personal COVID-19 preventive measures in the past month**			
Frequency of face mask wearing in public spaces or on transportation other than the workplace			
Every time	155 (74.5)	179 (78.2)	
Often	34 (16.3)	44 (19.2)	
Sometimes	16 (7.7)	5 (2.2)	
Never	3 (1.4)	1 (0.4)	0.03
Frequency of face mask wearing when you have close contact with other people in the workplace			
Every time	133 (63.9)	167 (72.9)	
Often	45 (21.6)	49 (21.4)	
Sometimes	25 (12.0)	11 (4.8)	
Never	5 (2.4)	2 (0.9)	0.02
Frequency of sanitizing of hands, using soaps, liquid soaps, or alcohol-based sanitizer, after returning from public spaces or touching public installations			
Every time	110 (52.9)	109 (47.6)	
Often	53 (25.5)	69 (30.1)	
Sometimes	41 (19.7)	44 (19.2)	
Never	4 (1.9)	7 (3.1)	0.57
Self-reported avoiding of social and meal gathering with other people who do not live together			
No	82 (39.4)	93 (40.6)	
Yes	126 (60.6)	136 (59.4)	0.88
Self-reported avoiding of crowded places			
No	72 (34.6)	81 (35.4)	
Yes	136 (65.4)	148 (64.6)	0.95

**Table 2 vaccines-11-00170-t002:** Changes in parental acceptability and attitudes related to children’s COVID-19 vaccination.

	Round 1 (*n* = 208)	Round 2 (*n* = 229)	Unadjusted *p* Values	Adjusted *p* Values ^a^	Effect Size
	*n* (%)	*n* (%)			
**Parental acceptability of COVID-19 vaccination for their child aged 6–35 months**					
Likelihood of having the child take up COVID-19 vaccination					
Very unlikely/unlikely/neutral	69 (33.2)	47 (20.5)			
Likely/very likely	139 (66.8)	182 (79.5)	0.004	0.01	0.29
**Parental hesitancy to receive COVID-19 vaccination**					
No	171 (82.2)	228 (99.6)			
Yes	37 (17.8)	1 (0.4)	<0.001	<0.001	0.74
**Attitudes toward COVID-19 vaccination for their children**					
Positive attitudes toward COVID-19 vaccination, *n* (%) agree					
COVID-19 vaccination is highly effective in protecting your child from COVID-19	119 (57.2)	165 (72.1)	0.002	0.02	0.31
Taking up COVID-19 vaccination can contribute to the control of COVID-19 in China	180 (86.5)	214 (93.4)	0.02	0.01	0.23
China will have an adequate supply of COVID-19 vaccine	161 (77.4)	204 (89.1)	0.002	0.001	0.32
Positive Attitude Scale score, mean (SD)	8.1 (1.1)	8.5 (0.9)	<0.001	0.002	0.34
Negative attitudes toward COVID-19 vaccination, *n* (%) agree					
Your child will have severe side effects after receiving COVID-19 vaccination	27 (13.0)	33 (14.4)	0.77	0.89	0.04
The protection of COVID-19 vaccines will only last for a short time	45 (21.6)	114 (49.8)	<0.001	<0.001	0.60
Your child is afraid of vaccination	49 (23.6)	55 (24.0)	1.00	0.62	0.009
You do not have time to take your child for COVID-19 vaccination	33 (15.9)	57 (24.9)	0.03	0.04	0.22
Negative Attitude Scale score, mean (SD)	7.6 (1.6)	7.9 (1.9)	0.06	0.03	0.18
Perceived subjective norm related to child’s COVID-19 vaccination: your family member would support you in having the child take up COVID-19 vaccination					
Agree, *n* (%)	105 (50.5)	156 (68.1)	<0.001	<0.001	0.36
Response score, mean (SD)	2.4 (0.6)	2.6 (0.6)	<0.001	0.004	0.33
Perceived behavioural control to have the child take up COVID-19 vaccination: having the child receive COVID-19 vaccination is easy for you if you want them to					
Agree, *n* (%)	99 (47.6)	143 (62.4)	0.003	0.02	0.30
Response score, mean (SD)	2.4 (0.6)	2.6 (0.6)	0.002	0.01	0.29
**Frequency of exposing to the following information on social media (e.g., WeChat, WeChat moments, Weibo, Tiktok) in the past month**					
Testimonials given by recipients of COVID-19 vaccination on social media					
Almost none	100 (48.1)	53 (23.1)			
Seldom	55 (26.4)	72 (31.4)			
Sometimes	33 (15.9)	69 (30.1)			
Always	20 (9.6)	35 (15.3)	<0.001	<0.001	0.34
Response score, mean (SD)	1.9 (1.0)	2.4 (1.0)	<0.001	<0.001	0.50
COVID-19 pandemic is not under control in some countries after scaling up COVID-19 vaccination					
Almost none	N.A.	39 (17.0)			
Seldom	N.A.	79 (34.5)			
Sometimes	N.A.	66 (28.8)			
Always	N.A.	45 (19.7)			
Response score, mean (SD)	N.A.	2.5 (1.0)	N.A.	N.A.	N.A.
Infectiousness and harms of the variants concern of COVID-19					
Almost none	N.A.	26 (11.4)			
Seldom	N.A.	53 (23.1)			
Sometimes	N.A.	88 (38.4)			
Always	N.A.	62 (27.1)			
Response score, mean (SD)	N.A.	2.8 (1.0)	N.A.	N.A.	N.A.
Outbreak caused by variants concern of COVID-19 in some places of China					
Almost none	N.A.	34 (14.8)			
Seldom	N.A.	71 (31.0)			
Sometimes	N.A.	82 (35.8)			
Always	N.A.	42 (18.3)			
Response score, mean (SD)	N.A.	2.6 (1.0)	N.A.	N.A.	N.A.
People contract COVID-19 after receiving primary series of COVID-19					
Almost none	N.A.	50 (21.8)			
Seldom	N.A.	93 (40.6)			
Sometimes	N.A.	71 (31.0)			
Always	N.A.	15 (6.6)			
Response score, mean (SD)	N.A.	2.2 (0.9)	N.A.	N.A.	N.A.

^a^: Adjusted *p* values: adjusted for age of the parent, education level, frequency of face mask wearing in public spaces or on transportation other than the workplace, and frequency of face mask wearing when you have close contact with other people in the workplace.

**Table 3 vaccines-11-00170-t003:** Associations between background characteristics and parental acceptability of COVID-19 vaccination for their children aged 6–35 months.

	Round 1 (*n* = 208)		Round 2 (*n* = 229)	
	OR (95%CI)	*p* Values	OR (95%CI)	*p* Values
**Sociodemographic characteristics**				
Age group, years				
18–30	1.0		1.0	
31–40	1.57 (0.87, 2.84)	0.14	0.88 (0.45, 1.74)	0.72
>40	0.91 (0.15, 5.68)	0.92	6.48 (0.82, 51.37)	0.08
Sex				
Male	1.0		1.0	
Female	1.14 (0.64, 2.04)	0.66	2.64 (1.36, 5.11)	0.004
Relationship status				
Married	1.0		1.0	
Single or divorced	N.A.	N.A.	1.03 (0.11, 9.47)	0.98
Having a stable partner	0.50 (0.03, 8.18)	0.63	N.A.	N.A.
Education				
Senior high or below	1.0		1.0	
College and above	0.90 (0.50, 1.63)	0.73	0.36 (0.18, 0.71)	0.003
Monthly personal income, RMB (USD)				
<3000 (462)	1.0		1.0	
3000–6999 (462–1077)	1.17 (0.57, 2.38)	0.67	0.54 (0.17, 1.65)	0.28
≥7000 (1078)	0.94 (0.41, 2.16)	0.88	0.32 (0.10, 1.04)	0.06
Type of work				
Frontline workers	1.0		1.0	
Management staff	0.77 (0.43, 1.40)	0.39	0.63 (0.32, 1.23)	0.17
Age of the child, months				
24–35	1.0		1.0	
12–23	1.00 (0.56, 1.81)	0.99	1.00 (0.62, 1.93)	0.99
6–11	0.74 (0.12, 4.62)	0.75	1.73 (0.37, 8.18)	0.49
**Personal COVID-19 preventive measures in the past month**				
Frequency of face mask wearing in public spaces or on transportation other than the workplace				
Never/sometimes/often	1.0		1.0	
Every time	1.82 (0.96, 3.46)	0.07	1.50 (0.72, 3.12)	0.28
Frequency of face mask wearing when you have close contact with other people in the workplace				
Never/sometimes/often	1.0		1.0	
Every time	1.01 (0.56, 1.84)	0.97	3.12 (1.60, 6.12)	0.001
Frequency of sanitizing of hands, using soaps, liquid soaps, or alcohol-based sanitizer, after returning from public spaces or touching public installations				
Never/sometimes/often	1.0		1.0	
Every time	1.04 (0.59, 1.86)	0.89	2.92 (1.45, 5.89)	0.003
Self-reported avoiding of social and meal gathering with other people who do not live together				
No	1.0		1.0	
Yes	2.21 (1.22, 3.98)	0.01	0.99 (0.52, 1.90)	0.98
Self-reported avoiding of crowed places				
No	1.0		1.0	
Yes	2.14 (1.18, 3.90)	0.01	0.82 (0.42, 1.63)	0.58

OR: crude odds ratios. N.A.: not applicable.

**Table 4 vaccines-11-00170-t004:** Factors associated with parental acceptability of COVID-19 vaccination for their children aged 6–35 months.

	Round 1 (*n* = 208)				Round 2 (*n* = 229)			
	OR (95%CI)	*p* Values	AOR (95%CI)	*p* Values	OR (95%CI)	*p* Values	AOR (95%CI)	*p* Values
**Parental hesitancy to receive COVID-19 vaccination**								
No	1.0		1.0					
Yes	0.26 (0.18, 0.38)	<0.001	0.25 (0.12, 0.54)	<0.001	N.A.	N.A..	N.A.	N.A.
**Attitudes toward COVID-19 vaccination for their children**								
Positive Attitude Scale	1.88 (1.41, 2.50)	<0.001	1.86 (1.38, 2.50)	<0.001	2.04 (1.47, 2.85)	<0.001	1.99 (1.40, 2.85)	<0.001
Negative Attitude Scale	0.96 (0.80, 1.14)	0.61	0.96 (0.80, 1.16)	0.69	0.86 (0.72, 1.01)	0.07	0.80 (0.66, 0.97)	0.02
Perceived subjective norm	4.68 (2.66, 8.23)	<0.001	4.69 (2.64, 8.32)	<0.001	8.07 (4.20, 15.48)	<0.001	8.33 (4.12, 16.83)	<0.001
Perceived behavioural control	1.91 (1.21, 2.99)	0.005	2.02 (1.27, 3.21)	0.003	2.90 (1.73, 4.87)	<0.001	3.09 (1.76, 5.43)	<0.001
**Frequency of exposing to the following information on social media (e.g., WeChat, WeChat moments, Weibo, Tiktok) in the past month**								
Testimonials given by recipients of COVID-19 vaccination on social media	1.22 (0.91, 1.65)	0.19	1.24 (0.91, 1.68)	0.18	0.96 (0.70, 1.33)	0.83	0.79 (0.55, 1.13)	0.20
COVID-19 pandemic is not under control in some countries after scaling up COVID-19 vaccination	N.A.	N.A.	N.A.	N.A.	0.85 (0.61, 1.18)	0.32	0.82 (0.57, 1.17)	0.27
Infectiousness and harms of the variants concern of COVID-19	N.A.	N.A.	N.A.	N.A.	0.87 (0.62, 1.22)	0.41	0.78 (0.53, 1.14)	0.20
Outbreak caused by variants concern of COVID-19 in some places of China	N.A.	N.A.	N.A.	N.A.	0.77 (0.54, 1.08)	0.13	0.80 (0.55, 1.17)	0.25
People contract COVID-19 after receiving primary series of COVID-19	N.A.	N.A.	N.A.	N.A.	0.88 (0.61, 1.28)	0.50	0.92 (0.62, 1.38)	0.69

OR: crude odds ratios. AOR: adjusted odds ratios, odds ratios adjusted for significant background characteristics in Table 3. N.A.: not applicable.

## Data Availability

The data presented in this study are available from the corresponding author upon request. The data are not publicly available as they contain personal behaviours.

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
