# Peer review of "Changes in COVID-19 Vaccine Acceptability among Parents with Children Aged 6–35 Months in China—Repeated Cross-Sectional Surveys in 2020 and 2021"

_vaccines, 2023, doi:10.3390/vaccines11010170_

Round 1

Reviewer 1 Report

The article discusses parents' intention to potentially have their younger children vaccinated, should the COVID-19 vaccine be made freely available for children aged 3-36 months old. It makes contributions in at least two directions: 1) establishing a base parental acceptability and 2) observing the changes in parental acceptability from 2020 to 2021.

The paper is well written and is easy to follow.

The motivation for the work is established well in the introduction, with compelling statistics presented that point to the dangers faced by younger children and the need to take preventive action and the nice introduction of parental acceptability as a parameter. The research is designed well, the conclusions are supported by the data and the discussion is adequate. A number of weaknesses are identified and fairly discussed.

There are two weaknesses in the work. The first one is about relevance. With the pandemic changing rapidly, one has to wonder how reliable and relevant are a the end of 2022 the findings from the fall of 2021. Especially for China, the recent political change directing the country away from its earlier zero covid policy has created a fundamentally new landscape. This is not to say that the authors' research is not of value any more. But the weakness should at least be acknowledged and discussed. 

The second weakness is about novelty. It is not clear what the difference is between this paper and the authors' previously published work, particularly the one listed as [40] in the bibliography. The authors are urged to provide more information regarding how this manuscript is different than their earlier works that are based on the same data.

This last point is actually my main concern with this work. If this is clarified and it is shown that there is sufficient new content in this manuscript, then I would see no reason to delay its publication as it makes a valid contribution in a crucial field. 

Author Response

1. The article discusses parents' intention to potentially have their younger children vaccinated, should the COVID-19 vaccine be made freely available for children aged 3-36 months old. It makes contributions in at least two directions: 1) establishing a base parental acceptability and 2) observing the changes in parental acceptability from 2020 to 2021. The paper is well written and is easy to follow.

A: Thank you for your comments.

2. The motivation for the work is established well in the introduction, with compelling statistics presented that point to the dangers faced by younger children and the need to take preventive action and the nice introduction of parental acceptability as a parameter. The research is designed well, the conclusions are supported by the data and the discussion is adequate. A number of weaknesses are identified and fairly discussed.

A: Thank you for your comments

3. There are two weaknesses in the work. The first one is about relevance. With the pandemic changing rapidly, one has to wonder how reliable and relevant are at the end of 2022 the findings from the fall of 2021. Especially for China, the recent political change directing the country away from its earlier zero covid policy has created a fundamentally new landscape. This is not to say that the authors' research is not of value any more. But the weakness should at least be acknowledged and discussed. 

A: Similar to the time when this study was conducted, COVID-19 vaccination is not yet available for children aged 6-35 months in China by the end of 2022. We agreed with the reviewer that our findings are most applicable to the time when zero COVID policy was implemented in China. When China started to relieve its COVID-19 control measures, number of COVID-19 cases increased dramatically. Parents may perceive a much stronger need to vaccinate their children against COVID-19. We have acknowledged this weakness in the limitation session.

4. The second weakness is about novelty. It is not clear what the difference is between this paper and the authors' previously published work, particularly the one listed as [40] in the bibliography. The authors are urged to provide more information regarding how this manuscript is different than their earlier works that are based on the same data. This last point is actually my main concern with this work. If this is clarified and it is shown that there is sufficient new content in this manuscript, then I would see no reason to delay its publication as it makes a valid contribution in a crucial field. 

A: This paper has some differences when comparing to our previously published work. First, the study population is different. The previous published paper focused on parents with children aged 3-17 years, while this one targeted parents with children aged 6-35 months. The COVID-19 vaccination policy is very different between children aged 3-17 years and those aged 6-35 months in China. China recommended COVID-19 vaccination for children aged 3-17 years and started implementing national vaccination program for this group of children in July 2021 (before the second round of survey). However, the country has not yet recommended or offered COVID-19 vaccination to children aged 6-35 months. Second, the dependent variable is different. The published work used vaccine hesitancy as the dependent variable (yes=very unlikely/unlikely/neutral, and no=likely/very likely/already vaccinated). This paper use acceptability as the dependent variable (yes=likely/very likely, and no=very unlikely/unlikely/neutral). Third, previous studies suggested that level of parental acceptability and its determinants for older children and younger children might be different [1]. Given the different contexts (e.g., policy, availability of vaccination) between children aged 3-17 years and those aged 6-35 months, it is likely that determinants of vaccine acceptability/hesitancy identified by our previous work (targeting children aged 3-17 years) might not be applicable to younger children. According to the principle of social marketing, health promotion should tailor to the needs of the target population [2]. Therefore, it is necessary to look at determinants of parental acceptability of COVID-19 vaccination for children aged 6-35 months. This study was one of the first studies looking at this age group of children in China. We have elaborated the difference between this paper and our previously published work in the revised manuscript.

References

[1] Galanis P, et al. Willingness, refusal and influential factors of parents to vaccinate their children against the COVID-19: A systematic review and meta-analysis. Preventive Medicine. 2022, 157, 106994.

[2] Valente TW, Fosados R. Diffusion of innovations and network segmentation: the part played by people in promoting health. Sex Transmitted Diseases, 2006; 33: S23–31

Reviewer 2 Report

thank you for letting me review this interesting paper.

The paper is a research paper about  the changes in COVID-19 vaccine acceptability and associated factors among parents with children aged 6-35 months in 2020 and 2021 that is an hot topic worldwide.

Background is well written I just suggest to read and include in the references the following articles: DOI: 10.3390/vaccines9060538

  • DOI: 10.3390/vaccines9050500

and I suggest to compare in the discussion the Chinese experience to international experience.

Author Response

1. Thank you for letting me review this interesting paper. The paper is a research paper about the changes in COVID-19 vaccine acceptability and associated factors among parents with children aged 6-35 months in 2020 and 2021 that is a hot topic worldwide.

A: Thank you for your comments

2. Background is well written I just suggest to read and include in the references the following articles: DOI: 10.3390/vaccines9060538, DOI: 10.3390/vaccines9050500 and I suggest to compare in the discussion the Chinese experience to international experience.

A: Thank you for your suggestions. We have included the references suggested by the reviewer in the revised manuscript.

We have also elaborated the discussion by comparing Chinese experiences to international experience in the revised manuscript.

“As compared to some other jurisdictions (i.e., United States, Australia, Canada, and Hong Kong SAR of China) [1-4], mainland China was late in rolling out COVID-19 vaccination programs for children aged below three years. China may learn from experience and evidence obtained from these jurisdictions and expand the existing COVID-19 vaccination programs to children aged 6-35 months. This is especially important for protecting these young children after the country decided to change its zero COVID policy.”

References

[1] Administration, F.a.D. Pfizer-BioNTech COVID-19 vaccine EUA letter of authorization reissued July 8, 2022. Silver Spring, MD: US Department of Health and Human Services, Food and Drug Administration. 2022; Available from: https://www.fda.gov/media/150386/download.

[2] Australian Government, ATAGI recommend the COVID-19 vaccination for some children aged 6 months to under 5 years. Available at: https://www.health.gov.au/news/atagi-recommend-the-covid-19-vaccination-for-some-children-aged-6-months-to-under-5-years. Accessed on January 3, 2023.

[3] National Advisory Committee on Immunization (NACI), Recommendations on the use of Moderna Spikevax COVID-19 vaccine in children 6 months to 5 years of age. Available at: https://www.canada.ca/content/dam/phac-aspc/documents/services/immunization/national-advisory-committee-on-immunization-naci/recommendations-use-moderna-spikevax-covid-19-vaccine-children-6-months-5-years.pdf. Accessed on January 3, 2023.

[4] The Government of the Hong Kong SAR, COVID-19 vaccination of children aged under 3 begins today. Available at: https://www.info.gov.hk/gia/general/202208/04/P2022080400470.htm. Accessed on January 3, 2023

Reviewer 3 Report

Dear Editor, Dear Authors,

Thank You for the opportunity to review the manuscript entitled Changes in COVID-19 vaccine acceptability among parents with 2 children aged 6-35 months in China --- repeated cross-sectional 3 surveys in 2020 and 2021. The manuscript is well written and addresses the important topic of the COVID-19 vaccine acceptance.

To get better scientific soundness I suggest considering the following remarks: It is not clear who was really investigated in this study, as under the title the authors wrote “… among parents with 2 children aged …” but under 2.2 Participants and data collection they stated “a sub-sample of the participants with at least 97 one child aged 6-35 months were included”. Clarify and correct, please.

It is not clear how the difference in categorical variables presented in the table (2 which were probably compared in the unadjusted analyses by the chi-square test) were additionally adjusted for the other variables, clarify please.

It was not explained, and it is not clearly understandable to me, why authors omitted the association between positive and negative attitudes and the acceptance of child vaccination in the univariable logistic analyses (crude OR). I suggest adding those results to the table 2.

It would be worth knowing the AOR for the individual elements of positive and negative attitudes, as they were mentioned in the table 2, not for the scale only.

It is not clear how ‘parental hesitancy’ has been defined. Is this the same as being vaccinated? It is not clear whether this factor has been considered as a factor associated with the parental acceptability.

As in the sample there were also families with children older than 35 months which were vaccinated. The positive/negative vaccination experience in other children surely is a key determinant associated with the vaccine acceptability for younger children. I suggest adding those results to the content of the manuscript (especially as the authors highlight this issue under discussion).

As the study sample differs from the several regions (less developed in China, other worldwide), generalization is rather not possible, therefore I suggest adding a sentence describing the characteristic of the population studied at the end of the first sentence under Conclusions. 

Author Response

1. Dear Editor, Dear Authors,

Thank You for the opportunity to review the manuscript entitled Changes in COVID-19 vaccine acceptability among parents with 2 children aged 6-35 months in China --- repeated cross-sectional 3 surveys in 2020 and 2021. The manuscript is well written and addresses the important topic of the COVID-19 vaccine acceptance.

A: Thank you for your comments.

2. To get better scientific soundness I suggest considering the following remarks: It is not clear who was really investigated in this study, as under the title the authors wrote “… among parents with 2 children aged …” but under 2.2 Participants and data collection they stated “a sub-sample of the participants with at least 97 one child aged 6-35 months were included”. Clarify and correct, please.

A: Sorry for the confusion. We have clarified that our study population was parents having children aged 6-35 months. The original repeated cross-sectional surveys were conducted among factory workers with children aged under 18 years. We extracted the data of parents having children aged 6-35 months from the datasets.

3. It is not clear how the difference in categorical variables presented in the table (2 which were probably compared in the unadjusted analyses by the chi-square test) were additionally adjusted for the other variables, clarify please.

A: Sorry for the confusion. We have elaborated the details in the statistical analysis of the revised manuscript:

“Difference in background characteristics, parental acceptability, attitudes toward COVID-19 vaccination and social media influence between participants in the first and second round of survey were compared using chi-square tests (for categorical variables) or independent sample t tests (for continuous variables). Unadjusted p values were obtained. Using parental acceptability, attitudes toward COVID-19 vaccination and social media influence as dependent variables, round of survey (second round versus first round) as independent variable and adjusted for significant background characteristics with between-round differences, adjusted p values were obtained using logistic regression models (for binary dependent variables) or linear regression models (for continuous dependent variables).”

4. It was not explained, and it is not clearly understandable to me, why authors omitted the association between positive and negative attitudes and the acceptance of child vaccination in the univariable logistic analyses (crude OR). I suggest adding those results to the table 2.

A: We agreed with the reviewer and added crude OR between all independent variables of interest and the dependent variables in the revised Table 4.

5. It would be worth knowing the AOR for the individual elements of positive and negative attitudes, as they were mentioned in the table 2, not for the scale only.

A: We agreed with the reviewer and added crude OR and AOR for the associations between individual items of positive/negative attitudes and parental acceptability in the Appendix 1. We have also included the findings in the revised results and discussion.

6. It is not clear how ‘parental hesitancy’ has been defined. Is this the same as being vaccinated? It is not clear whether this factor has been considered as a factor associated with the parental acceptability.

A: Sorry for the confusion. Unvaccinated parents who were very unlikely/unlikely/neutral to take up COVID-19 vaccination in future were defined as having “parental vaccine hesitancy”. If the parents who have already vaccinated against COVID-19, or likely/very likely to receive COVID-19 vaccination in future, they were regarded as no parental vaccine hesitancy in this study. We have elaborated the definition in the revised method of the manuscript.

7. As in the sample there were also families with children older than 35 months which were vaccinated. The positive/negative vaccination experience in other children surely is a key determinant associated with the vaccine acceptability for younger children. I suggest adding those results to the content of the manuscript (especially as the authors highlight this issue under discussion).

A: In this study, participants referred to the one whose birthday was closest to the survey date when answering questions related to parental acceptability if they had more than one child under the age of 18 years. Therefore, the vaccination status for other children was not investigated by this study. We agreed with the reviewer that other children’s vaccination status was an important determinant for parental acceptability. We have acknowledged this point as one major limitation of this study.

8. As the study sample differs from the several regions (less developed in China, other worldwide), generalization is rather not possible, therefore I suggest adding a sentence describing the characteristic of the population studied at the end of the first sentence under Conclusions.

A: We agreed with the reviewer and added the following sentence to the conclusion:

“This study compared COVID-19 vaccine acceptability for children aged 6-35 months in 2020 and 2021 among factory workers in Shenzhen, which is one of the four most developed cities in China”.

Round 2

Reviewer 1 Report

Already from the previous round of reviewing it was clear that the manuscript was of high quality. My core concern at that time was a potential overlap with the authors' previously published work. 

The authors have now clarified that there is no such overlap - with one paper focusing on children above 3 years of age and the other on younger children and the two demographics following different vaccination guidance - and therefore I have no reason to recommend anything other than publication.

It is a very interesting work and it is presented nicely.

Author Response

Already from the previous round of reviewing it was clear that the manuscript was of high quality. My core concern at that time was a potential overlap with the authors' previously published work. 

The authors have now clarified that there is no such overlap - with one paper focusing on children above 3 years of age and the other on younger children and the two demographics following different vaccination guidance - and therefore I have no reason to recommend anything other than publication.

It is a very interesting work and it is presented nicely.

A: Thank you for your comments.

Reviewer 3 Report

Dear Editor, Dear Authors,

the Authors clarified all the issues properly. In my opinion the manuscript has been sufficiently improved to be published in Vaccines.

Reviewer

Author Response

Dear Editor, Dear Authors,

the Authors clarified all the issues properly. In my opinion the manuscript has been sufficiently improved to be published in Vaccines.

A: Thank you for your comments.